# Optimization of the Clinical Setting Using Numerical Simulations of the Electromagnetic Field in an Obese Patient Model for Deep Regional Hyperthermia of an 8 MHz Radiofrequency Capacitively Coupled Device in the Pelvis

**DOI:** 10.3390/cancers13050979

**Published:** 2021-02-26

**Authors:** Takayuki Ohguri, Kagayaki Kuroda, Katsuya Yahara, Sota Nakahara, Sho Kakinouchi, Hirohide Itamura, Takahiro Morisaki, Yukunori Korogi

**Affiliations:** 1Department of Therapeutic Radiology, University Hospital of Occupational and Environmental Health, Kitakyushu 807-8555, Japan; k-yahara@med.uoeh-u.ac.jp; 2School of Information Science and Technology, Tokai University, Hiratsuka 259-1292, Japan; kagayaki@keyaki.cc.u-tokai.ac.jp; 3Department of Radiology, University of Occupational and Environmental Health, Kitakyushu 807-8555, Japan; sota0909@gmail.com (S.N.); kakino365@gmail.com (S.K.); blvd.avenue@gmail.com (H.I.); takam1989@med.uoeh-u.ac.jp (T.M.); ykorogi@med.uoeh-u.ac.jp (Y.K.)

**Keywords:** hyperthermia, capacitive heating, regional, temperature, simulation

## Abstract

**Simple Summary:**

Several randomized controlled trials have shown that the concurrent use of hyperthermia with radiotherapy results in a significant increase in local control rates. However, in studies that analyzed thermal parameters, the radiosensitizing effect required a favorable increase in tumor temperature. A good temperature increase is more difficult to achieve in deep-seated tumors than in superficial tumors. In this study, the reduction of subcutaneous fat overheating, which is a weak point of the deep heating of a capacitively coupled heating system, and the optimization of temperature distribution in the deep regional hyperthermia in the pelvis were investigated using electromagnetic field numerical simulations. In conclusion, the use of overlay boluses, the salt solution concentration in the overlay boluses, and the intergluteal cleft bolus insertion were found to be important for optimizing the temperature distribution. Further studies with numerical simulations based on the patient’s body shape and tumor location are expected.

**Abstract:**

*Background:* The purpose of this study was to evaluate the effectiveness of the clinical setting for deep regional hyperthermia of an 8 MHz radiofrequency (RF) capacitively coupled device in the pelvis by using numerical simulations of the electromagnetic field. *Methods:* A three-dimensional patient model of cervical cancer of the uterus in an obese patient was reconstructed with computed tomography data. The specific absorption rate (SAR) and temperature distributions among the various heating settings were evaluated using numerical simulations. *Results:* The averaged SAR value of the deep target tumor was similar between with or without overlay boluses (OBs), and that of the subcutaneous fat (SF) at the edges of cooling boluses with OBs was lower than that of the SF without OBs. The use of OBs reduced the overheating of the SF. The 0.5% salt solution in the OB produced the least overheated areas outside the deep target tumor compared with the other concentrations. The insertion of the intergluteal cleft (IGC) bolus could improve the temperature concentration of the deep target tumor. *Conclusions:* The use of OBs and the salt solution concentration in the OB were important to optimize the temperature distribution. IGC bolus might contribute to temperature optimization. Further studies with individualized numerical simulations in each patient are expected.

## 1. Introduction

Hyperthermia is usually defined as a temperature elevation in the range of 39–45 °C, and temperatures beyond this range are considered thermal ablation [1]. Hyperthermia is known to have a direct cytotoxic effect on cancer cells while also acting as a radiosensitizer and chemosensitizer [1]. The efficacy of radiotherapy plus hyperthermia in various cancers was demonstrated and confirmed by randomized clinical trials [2,3,4]. In the experimental setting, substantial radiosensitization was observed at temperatures of ≥41 °C, and a clear thermal–dose effect relationship was observed in sensitization [5,6,7,8,9]. Numerous reports on superficial and deep-seated tumors treated with radiotherapy plus hyperthermia have indicated a positive interrelationship between thermal parameters and clinical outcomes [4,10,11,12]. However, in deep-seated tumors, the temperature can be often difficult to increase to >41.0 °C [13].

The use of regional heating for the deep area of the human body has been previously investigated, particularly for the treatment of pelvic tumors and soft-tissue sarcomas [4,14]. Deep regional heating systems can be classified into two types, radiative and capacitive types [15]. Many researchers indicate the ability of radiofrequency (RF)-capacitive systems to achieve regional deep heating [11,16,17,18]. When using this capacitive device, the patient is placed between two electrodes connected to a high-power RF generator. The overheating of the subcutaneous fat (SF) tissue and pain near the electrode edge are major problems of deep regional heating. Excessive power deposition in the fatty tissue limits the effectiveness of the capacitive technique. Asian patients are considered relatively suitable owing to their slender appearance.

The hyperthermia treatment planning for deep-seated tumors has been investigated, although there are uncertainties and limitations related to electrical parameters and real blood flow in various organs and tumors, as well as reproducibility of the applicators and electrodes of the heating system. In particular, for the radiative heating device, promising results were reported for simulating power absorption and/or temperature patterns, which help operators visualize the effects of different steering strategies in modern locoregional hyperthermia treatments [19,20]. An 8 MHz RF capacitive device has been widely used in Japan for a long time since the 1990s. Although various clinical settings for heating have been used, only a few reports have verified their effectiveness by numerical analysis based on human models [21,22]. The purpose of this study was to evaluate the effectiveness of the current clinical heating settings of an 8 MHz RF capacitive heating device for pelvic tumors using numerical simulations of the electromagnetic field based on a personalized patient model.

## 2. Materials and Methods

### 2.1. Patient Model

An obese patient (body mass index: 29) with cervical cancer of the uterus was used in this study. The three-dimensional (3D) patient model was reconstructed with computed tomography (CT) data sets composed of 67 slices and with a slice thickness of 7 mm. The CT images were taken with the OB attached to the skin surface of the patient’s pelvic region (Figure 1). In addition, the effectiveness of the prone position was reported as a heating position in capacitively coupled heating [23], and therefore, CT was imaged also in the prone position. This study was approved by our institutional review board.

The segmentations of each organ were performed using the medical image segmentation tool (iSEG, ZMT, Zurich, Switzerland). Muscle, fat, and bone were taken into account to generate the 3D patient model. Fat was also divided into subcutaneous and other fats. As a heating target, the primary tumor in the cervix was segmented.

### 2.2. Electromagnetic Models

The patient model in the simulation of this study was a constant factor, and the water bolus and its saline concentration were varied to study the impact on the specific absorption rate (SAR) and temperature. SAR distributions were computed using the low-frequency (LF) solver of SIM4LIFE version 3.2 (ZMT, Zurich, Switzerland) [24]. The LF solver used a finite-element method to analyze the spatial distributions of the electric scalar potential, whose gradient gives the electric field based on the quasistatic approximation of Maxwell’s equations. Then, SAR was calculated as a square of the electric field strength multiplied by electric conductivity and divided by the density of the medium. Because the electric conductivity varies strongly with the concentration of the saline, the resultant SAR distributions were different, with different bolus settings, even if the patient model was constant. The simulation setup was based on the clinically conventional setting for deep-seated tumors in pelvic regions by using an 8 MHz RF capacitive heating device [18]. Figure 1 shows the current conventional setting using the device for deep regional hyperthermia in the pelvic region. The diameters of both upper and lower electrodes were 30 cm. The liquid in the regular bolus (RB) adhering to the electrode and OB was a 5% salt solution set at 10 °C. Superficial cooling was performed using a circulating liquid of a 0.5% salt solution set at 5 °C in the OBs. An additional bolus of a 5% salt solution set at 10 °C was inserted in the intergluteal cleft (IGC) bolus to improve the temperature distribution of the deep target tumor.

Table 1 shows the electric conductivity (S/m), relative permittivity, and mass density (kg/m^3^) used for the simulations. These values were obtained from the IT’IS Foundation database. Because the frequency of the electric field was as low as 8 MHz and the electric conductivities in the tissues were not negligible compared with the dielectric permittivities, electro-quasistatic approximation was applied. The constant potential between the electrodes ranged from 40 to 70 V, according to various evaluated heating settings. The potential values were given as boundary conditions in the low-frequency solver. The size of each electromagnetic model of the SAR distribution ranged from 20.8 to 26.6 megacells.

Temperature distributions were simulated using the Pennes bioheat transfer equation-based transient-state solver until a steady state was reached [22,25,26]. All thermal models were created using the thermal solver of Sim4life. The bioheat transfer equation was described in previous articles [22,26], and the following parameters were used for each organ: specific heat capacity (J/kg K), thermal conductivity (W/m K), heat generation rate (W/kg), and perfusion rate (mL/min kg; Table 2). These values were also obtained from the IT’IS Foundation database. As boundary conditions in the simulations, the temperatures of the OBs were kept at a constant value of 5 °C and in ambient air at 25 °C. The size of each model of the temperature distribution ranged from 11.4 to 18.8 megacells.

The dielectric and thermal tissue properties of the deep target tumor in the cervical uterus were set to be identical with those of the muscle.

### 2.3. Evaluated Heating Settings

To assess the effect of the OBs, SAR and temperature distributions were generated with and without the use of the OBs; a constant potential of 55 V between the electrodes was used, and the simulation time was set at 3000 s, with a timestep of 1 s. In the settings with the OBs, a circulating liquid of a 0.5% salt solution set at 5 °C was used in the OBs. Without the OBs, a circulating liquid of a 0.5% salt solution set at 5 °C was used in the RBs. The deep target tumor/SF at the edges of the cooling boluses ratio for the averaged SAR value was defined as an index of heating efficiency; when with the OBs, the 2 cm area of SF at the edges of the OBs was used to calculate the averaged SAR value of the SF, and when without OBs, the 2 cm area of SF at the edges of the RBs was used.

To evaluate the effect of the concentration of the salt solution in the OBs, SAR and temperature distributions were simulated in the following heating settings, with the simulation time also set at 3000 s with a timestep of 1 s: (1) The constant potential between the electrodes ranged from 40 to 70 V at each concentration of the salt solution (0.1%, 0.5%, 1%, and 5%) of the circulating fluid in the OBs. (2) To investigate the effect of the concentration of the salt solution on the temperature distributions of the surrounding organs outside the deep target tumor, the value of the constant potentials between the electrodes was set to enable heating the deep target tumor to 40 °C according to the various concentrations of the salt solution (0.1%, 0.5%, 1%, and 5%) in the OBs.

In addition, the SAR and temperature distributions were also simulated to assess the effect of the IGC bolus (5% salt solution; Figure 1) when a 0.5% or 1% salt solution was used as the circulating fluid in the OBs.

## 3. Results

### 3.1. Effect of the OB

Figure 2 shows the effect of the use of the OBs on the SAR and temperature distributions between the settings with and without the OBs. The high-SAR areas in the SF were localized under the electrode in the setting without the OBs (Figure 2a), whereas those areas were widely dispersed in the setting with the OBs (Figure 2b). The high-SAR areas (yellow arrows) in the SF just outside the electrode were found in the setting without the OBs (Figure 2a). As these high-SAR areas are not expected to be cooled by the boluses, the appearance of pain due to overheating could be a concern. The averaged SAR value of the deep target tumor was 8.1 with OBs and 8.3 without OBs, and that of the SF at the edges of cooling boluses was 11.7 with OBs and 29.6 without OBs. The ratio of the deep target tumor to the SF at the edges of cooling boluses for the averaged SAR value was 0.69 with OBs, which was better than the 0.28 without OBs. For the temperature simulations, the overheating areas of the SF were clearly decreased in the setting with the use of the OBs in addition to the RBs compared with the setting without the OBs (Figure 2c,d).

### 3.2. Effect of the Salt Solution Concentration in the OB

Figure 3 shows the temperature graph of the axial cross-section of the deep target tumor center (Slice d of Figure 4) when the constant potential between the electrodes ranged from 40 to 70 V at each salt solution concentration in the OBs. The figure demonstrates that the salt solution concentration in the OB has a significant effect on the temperature increase in the body when heated by the same constant potential between electrodes.

Figure 4 indicates that the effect of the salt solution concentration in the OBs on the temperature distributions of the surrounding organs outside the deep target tumor when each constant potential between the electrodes was set to be able to heat the deep target to 40 °C. The first four vertical columns in Figure 4 show the changes in the temperature increase in the surrounding organs when the salt solution concentration in the OB ranged from 0.1% to 5%. The salt solution of 0.1% in the OBs caused overheating areas in the superficial region under the electrode, whereas the 5% or 1% salt solution caused overheating areas in the body outside the electrode. When the 0.5% salt solution in the OBs was used, the smallest overheated areas were found in the surrounding organs.

Figure 5 shows the temperature graphs for each slice section; overheated areas exceeding 44 °C in the body outside the electrodes were observed in slices d, e, or f when the 5% or 1% salt solution in the OBs was used. In slices b and f, overheated areas exceeding 44 °C in the SF under the electrode were observed when the 0.1% salt solution was used. When the 0.5% salt solution in the OBs was used, overheated areas exceeding 44 °C were not observed in any slice.

### 3.3. Effect of the IGC Bolus

The last two vertical columns of Figure 4 show the temperature distributions when an additional bolus (IGC bolus of the 5% salt solution) was inserted in the IGC when the 0.5% or 1% salt solution was used as the circulating fluid in the OB. Although a temperature increase zone was observed under the IGC, the temperature of the deep target tumor was increased, and the overheated SF, which might induce pain, was almost not observed.

Figure 6 shows the temperature graphs for each slice section and the effect of the IGC bolus on the temperature distribution when the 0.5% or 1% salt solution in the OB was used. A temperature increase of approximately 1 °C in the deep target tumor was obtained compared with the nonuse of the IGC bolus, and the overheated area of the surrounding organs was kept at <45 °C, although only in the ventral SF area contralateral to the IGC bolus insertion site (Slice b), where the temperature slightly exceeded 45 °C when the 0.5% salt solution in the OB was used.

## 4. Discussion

This is the first report to show that the effectiveness of the current clinical heating settings [18] such as the use of the OB and salt solution concentration of 0.5% in the OB of the 8 MHz RF capacitive heating device for pelvic tumors in the obese patient model was verified by numerical simulations of the electromagnetic field. In addition, the effect of the IGC bolus to optimize the temperature distribution was demonstrated.

A well-known disadvantage of a RF capacitive device for deep-seated tumors is the overheating of the SF tissue, especially at the electrode edges. The effectiveness of the OB in reducing the overheating that occurs at the electrode edges was reported in phantom experiments and clinical results [27,28]. In the present study, which was based on the numerical electromagnetic field simulations in obese patients, when the OBs were used, the averaged SAR values and temperature were clearly decreased in the SF at the electrode edges and under the electrodes without decreasing the averaged SAR value of the deep target tumor. Therefore, the use of OB could result in a good temperature increase in the deep target tumor by reducing pain due to overheating of the SF.

A few previous phantom studies indicated that optimization for the concentration of the salt solution in the boluses in contact with skin surface was one of the important factors to achieve a higher temperature increase in the deep heating target in the 8 MHz RF capacitive heating device [27,29]. Kato et al. reported a numerical analysis study of SAR distributions using an elliptical phantom for this heating device and showed the importance of optimizing the concentration of the salt solution in the boluses for deep heating [29]. When the salt solution concentration in the OBs was high, hot spots were generated in the fat layer near the edge of the OBs. On the other hand, when the salt solution concentration in the OBs was low, more RF current flowed in the SF layer under the electrodes, resulting in a higher SAR value. In the present study, even the temperature distribution was evaluated using a patient model, and the salt solution concentrations in the OBs were important to optimize the temperature distributions. The salt solution concentration of 0.5% in the OBs was estimated to be appropriate. The results were consistent with the phantom study for SAR distributions by Kato et al. In addition, anatomical overheated sites in the obese patient model were identified according to the salt solution concentrations in the OBs.

As shown in Figure 1, the OB does not adhere to the IGC in the conventional deep-heating setting. As many deep target tumors in the uterus, prostate, and rectum are located just below the IGC, the temperature increase of the deep target might be suppressed owing to the decrease in the RF current through the IGC. Therefore, deep heating with a 5% salt solution bolus (e.g., gauze soaked in the salt solution) inserted in the IGC was clinically performed in our hospital. The present study suggests that the placement of the IGC bolus improved the temperature concentration of the deep target tumor in the pelvis. As a mechanism for the results, almost no SF was found in the IGC, although both sides of the IGC were sandwiched by buttock fat. The insertion of a bolus of the 5% salt solution, which has high electrical conductivity, in the IGC area could allow the RF current to concentrate to the IGC with reduction of RF current to the buttock fat and could improve the temperature increase in the deep target tumor just below the IGC. Our group previously reported that the heating technique of inserting insulator sheets between the RB and OB and moving them temporally could improve the temperature increase for the prostate cancer of the deep pelvic tumor [30]. However, the weakness of the heating method using mobile insulating sheets is that it requires rotations of the insulating sheets temporally, which is time-consuming. IGC bolus insertion may be promising because it is more convenient and easier to implement.

Hyperthermia treatment planning (HTP) using numerical analyses of the electromagnetic field for the radiative heating device has recently been investigated to simulate temperature patterns and SAR distributions and help operators visualize the effects of different steering strategies in modern locoregional RF hyperthermia treatments [19,20]. Further investigations of HTP to optimize the heating settings in the individual patients based on each patient’s body shape and the location of the deep-seated tumor are expected for capacitively coupled heating.

A negative correlation between the subcutaneous fat thickness and deep temperature increase has been reported for the 8 MHz capacitive heating device [18,31,32]. In the present results for an obese patient, a 4–5 °C difference between deep tumor temperature and SF temperature was partially observed even with optimization of electrical conductivity in the boluses and use of the OBs and IGC bolus. Further clinical studies using these optimized methods for deep heating should be conducted for the patients with enough minimization of the temperature increase of the SF on the HTPs. In addition, the pathological significance and clinical symptoms of partial overheating of the SF at around 45 °C should be researched in particular.

The simulation of SAR and temperature distribution for hyperthermia using a human body model has some uncertainties and limitations as follows: uncertainties in the electric parameters and real blood flux and the contouring of the different tissues and the reproducibility of applicators and electrodes of the heating system. Therefore, our results can yield qualitative results that can serve as hints and educational illustrations but not as exact values. A further limitation of the present study was that the dielectric and thermal tissue properties of the tumor were calculated as equivalent to those of the muscle. Crezee et al. investigated the electrical conductivity of tumors in nine patients with cervical cancer by using the magnetic-resonance-based electrical properties of tomography [33]. The electrical conductivity of cervical cancer was variable among the patients but was relatively close to that of the muscle. Therefore, in the present study, the dielectric and thermal tissue properties for the deep target tumor were substituted with those for the muscle.

## 5. Conclusions

This is the first report to evaluate the effectiveness of the clinical setting for the deep regional heating of an 8 MHz RF capacitively coupled heating device in the pelvis by using numerical simulations of the electromagnetic field in the obese patient model. The use of OBs could reduce the overheating of the SF under and around the electrodes. The salt solution concentration in the OB was important to achieve the optimization of the deep-heating area. The salt solution of 0.5% in the OB was considered the clinical setting that produced the least overheated areas outside the deep target tumor. The insertion of the IGC bolus of the 5% salt solution could concentrate the RF current to the IGC area and improve the temperature concentration to the deep target tumor just below the IGC, although a temperature increase in the SF occurred partially. Further studies with individualized electromagnetic field simulations based on the patient’s body shape and tumor location are expected for the capacitively coupled heating device to optimize the temperature distribution in deep regional hyperthermia.

## Figures and Tables

**Figure 1 cancers-13-00979-f001:**
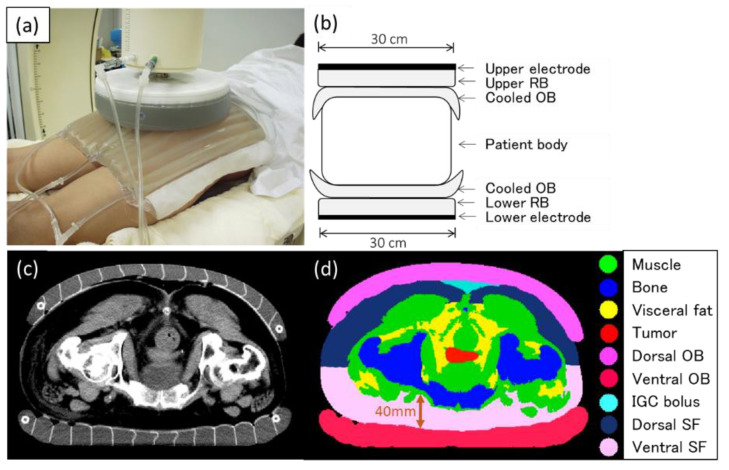
Conventional setting of deep regional hyperthermia for the pelvic region using the 8 MHz radiofrequency capacitive heating device. (**a**) Regular bolus, which attaches to electrodes of 30 cm in diameter, and overlay bolus (OB), in which internal cooled fluid circulates and cools the skin surface broadly. (**b**) Schematic diagram of deep regional hyperthermia using this device in the current conventional setting. RB, regular bolus; OB, overlay bolus. (**c**) Computed tomography (CT) image with the OBs in the prone position as in the heating position. (**d**) CT image after the segmentation of each organ. IGC, intergluteal cleft, SF, subcutaneous fat.

**Figure 2 cancers-13-00979-f002:**
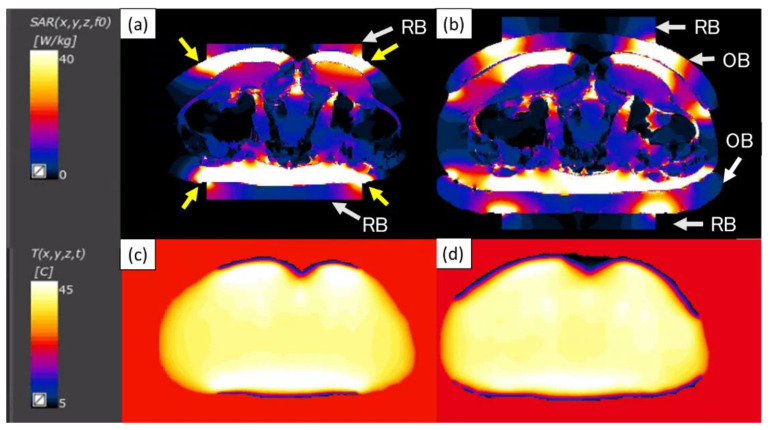
Specific absorption rate (SAR) distributions for the settings without (**a**) and with the overlay boluses (OBs) (**b**). The high-SAR areas ((**a**), yellow arrows) in the subcutaneous fat just outside the electrode were noticeable in the setting without the OBs. The temperature distributions for the settings with surface cooling by the circulating fluid in RBs without OBs (**c**) or the OBs (**d**) are shown. The overheated areas of the subcutaneous fat are clearly reduced by using the OBs (**d**) compared with the setting without the OBs (**c**). RB, regular bolus; OB, overlay bolus.

**Figure 3 cancers-13-00979-f003:**
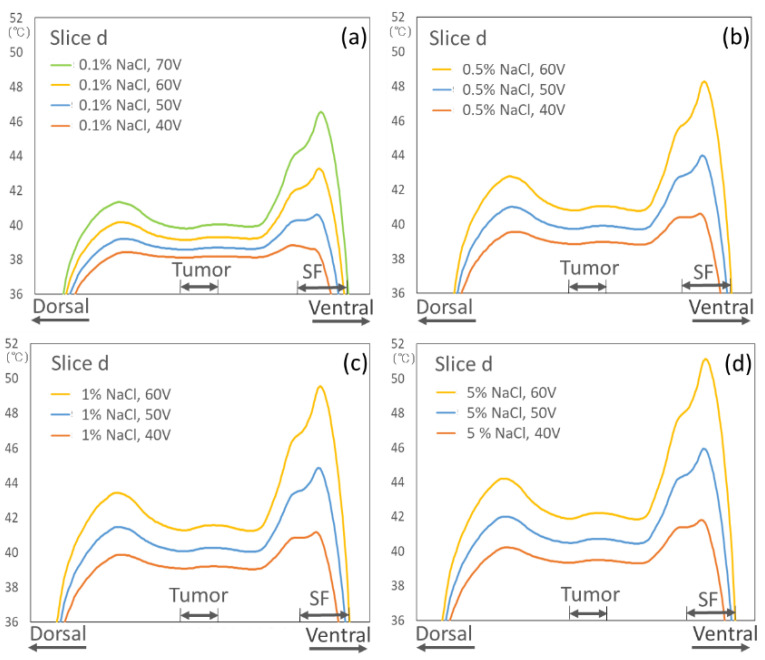
Temperature distributions in the body according to the constant potential between the electrodes at each salt solution concentration ((**a**) 0.1%, (**b**) 0.5%, (**c**) 1%, and (**d**) 5%) in the OBs.

**Figure 4 cancers-13-00979-f004:**
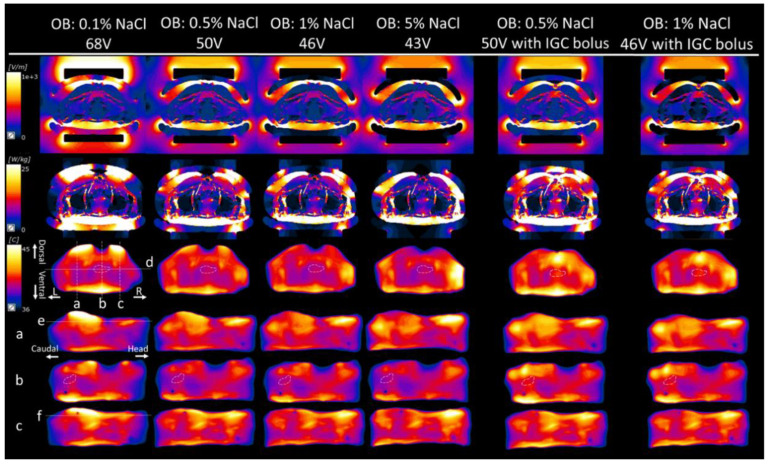
Effect of the salt solution concentration in the OBs on the temperature distributions of the surrounding organs outside the deep target tumor when each constant potential between the electrodes was set to be able to heat the deep target (white dotted line in the third vertical row) to 40 °C. Starting from the top row, the following are shown: electric field strength (*V*/*m*), specific absorption rate (SAR), and temperature distributions for the axial cross-sectional images of the deep target tumor center in the pelvis, followed by the sagittal cross-sectional image of line a, sagittal cross-sectional image of line b, and sagittal cross-sectional image of line c. The last two vertical columns are the electric field strength, SAR, and temperature distributions when an additional bolus (IGC bolus) was inserted in the IGC when the 0.5% or 1% salt solution was used as the circulating fluid in the OBs. d: line of a coronal section through the center of the deep target tumor, e: line of a coronal section through the dorsal subcutaneous fat on the sagittal cross-sectional image of line a, f: line of a coronal section through the dorsal subcuta-neous fat on the sagittal cross-sectional image of line c, L: left, R: right.

**Figure 5 cancers-13-00979-f005:**
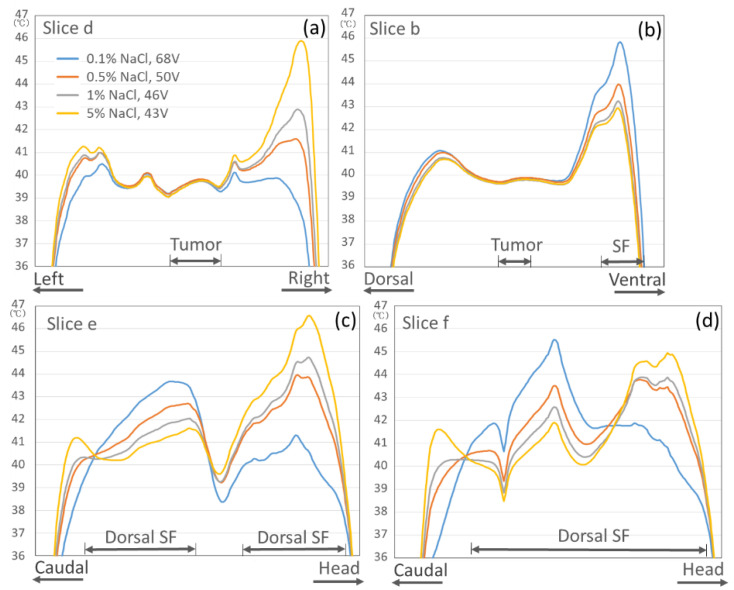
Effect of the salt solution concentration in the OBs on the temperature graphs for each slice section ((**a**) Slice d, (**b**) Slice b, (**c**) Slice e, and (**d**) Slice f) of the surrounding organs when each constant potential between the electrodes is set to be able to heat the deep target to 40 °C.

**Figure 6 cancers-13-00979-f006:**
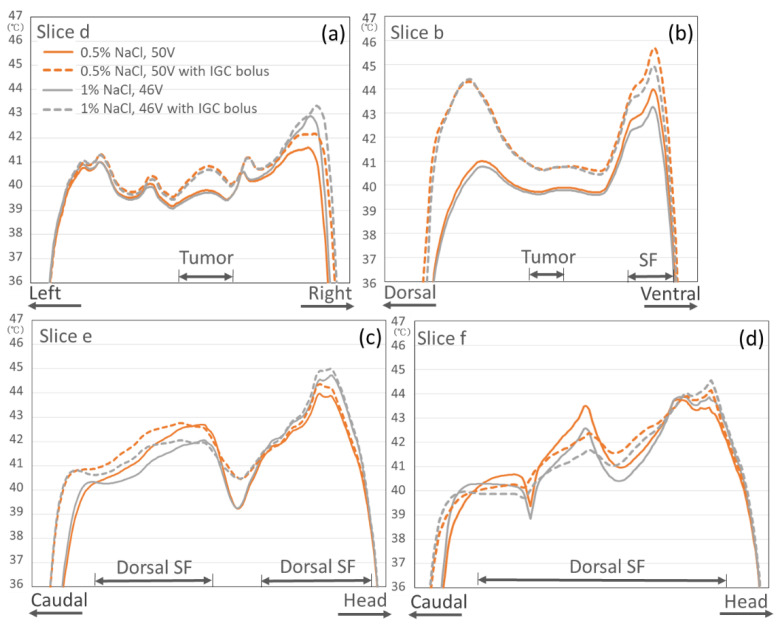
Effect of the additional bolus (IGC bolus of the 5% salt solution). The temperature graphs for each slice section ((**a**) Slice d, (**b**) Slice b, (**c**) Slice e, and (**d**) Slice f) compared with the nonuse of the IGC bolus when the 0.5% or 1% salt solution is used as the circulating fluid in the OBs.

**Table 1 cancers-13-00979-t001:** Dielectric and materials properties used for the simulations at 8 MHz.

Materials	Electric Conductivity (S/m)	Relative Permittivity	Mass Density (kg/m^3^)
Muscle	0.61	203	1090
Fat	0.05	33	911
Bone	0.04	43	1908
Regular boluses			
5% NaCl at 10 °C	6.31	82	1000
Overlay boluses			
0.1% NaCl at 5 °C	0.11	85	1000
0.5% NaCl at 5 °C	0.57	85	1000
1% NaCl at 5 °C	1.10	85	1000
5% NaCl at 5 °C	5.58	84	1000
IGC bolus			
5% NaCl at 10 °C	6.31	82	1000

IGC: intergluteal cleft.

**Table 2 cancers-13-00979-t002:** Thermal tissues properties used for the simulations at 8 MHz.

Materials	Specific Heat Capacity (J/kg K)	Thermal Conductivity (W/m K)	Heat Generation Rate (W/kg)	Perfusion Rate (mL/min kg)
Muscle	3421	0.49	0.91	36.7
Fat	2348	0.21	0.51	32.7
Bone	1312	0.32	0.15	10.0
IGC bolus	4180	6	0	0

## Data Availability

The data presented in this study are available on request from the corresponding author. The data are not publicly available due to privacy issues.

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
