# Peer review of "Optimization of the Clinical Setting Using Numerical Simulations of the Electromagnetic Field in an Obese Patient Model for Deep Regional Hyperthermia of an 8 MHz Radiofrequency Capacitively Coupled Device in the Pelvis"

_cancers, 2021, doi:10.3390/cancers13050979_

Round 1
Reviewer 1 Report
This paper present an application of hyperthermia using NaCl at different concentrations based in computer simulations. But unfortunately it is impossible to asses the results as it stands. My advice is to rewrite the paper explaining in detail the theoretical values for such salts in the body or to make any kind of measurements.
Author Response
Comment #1
This paper present an application of hyperthermia using NaCl at different concentrations based in computer simulations. But unfortunately it is impossible to asses the results as it stands. My advice is to rewrite the paper explaining in detail the theoretical values for such salts in the body or to make any kind of measurements.
Our reply
As the reviewer suggested, the computer simulations for hyperthermia using a human body model has some uncertainties and limitations. We added the sentences for the uncertainties, limitations and future clinical studies in the Discussion section.
Reviewer 2 Report
Comments
The RF8 capacitive hyperthermia device has been widely used in Japan. The technique of strong superficial cooling in the regular bolus and the use of overlay bolus has been quite a while. There has no specific paper explain why there is difference in the salt concentration between different bolus and why it should be. The manuscript is interesting in researchers of this field, because the efficacy of deep heating and overheat over subcutaneous fat has been a long existing problem of hyperthermia regardless the type of machine brands. The problems have been pointed out long ago but the solutions have not been perfect.
The authors have found the overlay bolus with different salt concentrations contribute to temperature optimization. They also found inserting an intergluteal cleft (IGC) bolus with high salt concentration can partially improve the deep seated tumor heating.
Suggestions
The temperature distributions were totally relied on computer assisted software based on theoretical models. However, it may not be correct in clinical validation. We suggest an intra-rectal and some skin temperature measurements should be performed with or without IGC bolus with different salt concentrations in at least one patient to prove the concept might be right.
Author Response
Comment #1
The temperature distributions were totally relied on computer assisted software based on theoretical models. However, it may not be correct in clinical validation. We suggest an intra-rectal and some skin temperature measurements should be performed with or without IGC bolus with different salt concentrations in at least one patient to prove the concept might be right.
Our reply
We agree with the reviewer's comments. Unfortunately, we cannot add the comparison among the patients for the IGC bolus and different salt concentrations, because the control patient is not readily available. We added the sentences for the future clinical studies in the Discussion section.
Reviewer 3 Report
The authors show interesting results of simulations and the influence of different parameters on the temperature distribution in hyperthermia by capacitive systems. Such calculations and estimations are important to see the applicability, the possibilities of optimizations and the limitations of these systems.
General comments:
- It is important to notice that simulations in HT show partly relevant deviations from measurements caused by uncertainties and limitations. These are e.g. uncertainties in the electric parameters, the real blood flux, the contouring of the different tissues and the exact modelling of the (individual) applicators/electrodes. Additionally, some of these values are changing during a single treatment or during the sequence of treatments. Therefore, such simulations can only yield qualitative results that can serve as hints and educational illustrations but not as exact values.
For instance, in the article the diagrams show a left-right asymmetry that is not expected. It could be the result of a slight asymmetric positioning of the patient in the CT or an asymmetric position of the electrodes.
The authors should respect this aspect of uncertainties by a paragraph in the introduction (or materials and methods chapter) and in the discussion. - The results show the principle limitation of the capacitive hyperthermia systems for thick patients. For all the efforts in optimization and strong surface cooling, the results show a large temperature difference of 4-5 °C between the target and the SF region that can yield either to too small temperatures in the tumour or to too large temperatures in the SF. The authors should discuss this.
- The advantage of the IGC bolus is not so clear. With and without the IGC bolus the limiting ratio of target to SF temperature (39,9/44 and 45,7/40,8 in slice b, fig. 6) is similar where the dorsal SF temperature is even increased with the IGC bolus in the IGC region. Maybe, other parts of the temperature distribution are improved by the better coupling of the E-field to the patient caused by the additional IGC bolus.
Further remarks:
- The authors label the different slices partly by letters and partly by roman numbers. Perhaps, it is less confusing to use only one labelling scheme.
- In fig. 1, there is no label SF in c but in d.
- In fig. 2, the left and the right side could be exchanged. Then the first/left side gives the results before and the second/right side the results after the addition of the OBs.
- In fig. 2 c and d, the difference are not clear seen (in both cases are large white regions). Maybe, a change in the scale can clarify this.
- In fig. 3, the diagrams have the label “slice b”. However, the diagrams show the results of slice I.
Author Response
Comment #1
It is important to notice that simulations in HT show partly relevant deviations from measurements caused by uncertainties and limitations. These are e.g. uncertainties in the electric parameters, the real blood flux, the contouring of the different tissues and the exact modelling of the (individual) applicators/electrodes. Additionally, some of these values are changing during a single treatment or during the sequence of treatments. Therefore, such simulations can only yield qualitative results that can serve as hints and educational illustrations but not as exact values.
For instance, in the article the diagrams show a left-right asymmetry that is not expected. It could be the result of a slight asymmetric positioning of the patient in the CT or an asymmetric position of the electrodes. The authors should respect this aspect of uncertainties by a paragraph in the introduction (or materials and methods chapter) and in the discussion.
Our reply
We agree with the reviewer's important suggestions, and added the sentences for uncertainties and limitations in the Introduction and Discussion sections.
Comment #2
The results show the principle limitation of the capacitive hyperthermia systems for thick patients. For all the efforts in optimization and strong surface cooling, the results show a large temperature difference of 4-5 °C between the target and the SF region that can yield either to too small temperatures in the tumour or to too large temperatures in the SF. The authors should discuss this.
Our reply
We added the sentences for the principle limitation of the capacitive hyperthermia systems for thick patients in the Discussion section on Page 9.
Comment #3
The advantage of the IGC bolus is not so clear. With and without the IGC bolus the limiting ratio of target to SF temperature (39,9/44 and 45,7/40,8 in slice b, fig. 6) is similar where the dorsal SF temperature is even increased with the IGC bolus in the IGC region. Maybe, other parts of the temperature distribution are improved by the better coupling of the E-field to the patient caused by the additional IGC bolus.
Our reply
As the reviewer suggested, when the IGC bolus is used, a temperature increase in SF is observed in slice b. This point is described in the Results section on page 8. The descriptions for the efficacy of the IGC bolus were modified in the Conclusion section.
Comment #4
The authors label the different slices partly by letters and partly by roman numbers. Perhaps, it is less confusing to use only one labelling scheme.
Our reply
As the reviewer suggested, the slice labels were unified with roman numbers.
Comment #5
In fig. 1, there is no label SF in c but in d.
Our reply
We revised in fig. 1 legend.
Comment #6
In fig. 2, the left and the right side could be exchanged. Then the first/left side gives the results before and the second/right side the results after the addition of the OBs.
Our reply
As the reviewer suggested, the position of the slide was changed in the Fig. 2, and revised the sentences in the Results section and Figure 2 legend.
Comment #7
In fig. 2 c and d, the difference are not clear seen (in both cases are large white regions). Maybe, a change in the scale can clarify this.
Our reply
The scale could not be changed properly. As a result, no changes were made.
Comment #8
In fig. 3, the diagrams have the label “slice b”. However, the diagrams show the results of slice I.
Our reply
As the reviewer suggested, we revised the fig. 3.
Round 2
Reviewer 1 Report
They have changed very few things from the first version and this paper needs much more explanation. For instance, the salt ClNa is in many parts of the body and it would be important to know how they calculate the SAR for the tumor.
Author Response
According to the reviewer comments for the calculation of SAR and patient model, we have made corrections in the Material and Methods section (2.2. Electromagnetic models).
Reviewer 2 Report
I accept the description about how current limitations related to SAR distributions computed from simulation model and further clinical studies should be validated in the future in the revised manuscript.
Author Response
I agree with the reviewer's helpful comments.
Round 3
Reviewer 1 Report
From my point of view, the authors didn't change anything and only advising the difficulties. Unfortunately this is not enough for a reader who is going to read this paper and to be sure that these study has an scientific worth. It is necessary to rewrite the paper taking into account realistic values with respet to a salt as the ClNa in a given tumor and to control the different distributions of heat, etc...